# TOP-DOWN TRAINING FOR NEURAL NETWORKS

## ABSTRACT

Vanishing gradients pose a challenge when training deep neural networks, resulting in the top layers (closer to the output) in the network learning faster when compared with lower layers closer to the input. Interpreting the top layers as a classifier and the lower layers a feature extractor, one can hypothesize that unwanted network convergence may occur when the classifier has overfit with respect to the feature extractor. This can lead to the feature extractor being under-trained, possibly failing to learn much about the patterns in the input data. To address this we propose a *good classifier hypothesis*: given a fixed classifier that partitions the space well, the feature extractor can be further trained to fit that classifier and learn the data patterns well. This alleviates the problem of under-training the feature extractor and enables the network to learn patterns in the data with small partial derivatives. We verify this hypothesis empirically and propose a novel top-down training method. We train all layers jointly, obtaining a good classifier from the top layers, which are then frozen. Following re-initialization, we retrain the bottom layers with respect to the frozen classifier. Applying this approach to a set of speech recognition experiments using the Wall Street Journal and noisy CHiME-4 datasets we observe substantial accuracy gains. When combined with dropout, our method enables connectionist temporal classification (CTC) models to outperform joint CTC-attention models, which have more capacity and flexibility.

## 1 INTRODUCTION

Deep neural networks (DNNs) have given impressive results in many domains, such as speech recognition (Hinton et al., 2012), natural language processing (Bahdanau et al., 2014) and computer vision (Krizhevsky et al., 2012); however, gradient-based training of DNNs remains a non-trivial task, owing to the vanishing gradient problem (Bengio et al., 1994; Hochreiter, 1998) which arises from the deep model structure.

The vanishing gradient problem does not only make the training slow, it can also result in the bottom layers closer to the input being under-trained, while over-training the top layers closer to the output. We can interpret the bottom layers of the neural network as the feature extractor, and the top layers as the classifier. During back-propagation, the gradient is decided by the current values of the parameters and all layers are trained jointly. Thus, the feature extractor is learning to provide features which fit the partition of the space defined by the current classifier, while the classifier is learning to fit the features generated by the current feature extractor.

Ideally, the speed of learning for the feature extractor and the classifier should be similar. However, since the partial derivatives of the top layers are much larger than the partial derivatives of the bottom layers, the classifier learns to fit to the feature extractor much faster than the feature extractor learns to fit to the classifier. Thus, the feature extractor layers are usually "under-trained" (they underfit the classifier) while the classifier are in general "over-trained" (they overfit the feature extractor). Also, since the feature extractor learns more slowly, then potentially the classifier may overfit the feature extractor before the feature extractor is able to learn much about the underlying pattern of the data.

Pre-training techniques can alleviate this problem (Hinton et al., 2006; Bengio et al., 2007; Larochelle et al., 2009; Erhan et al., 2010). In these methods, each layer is pre-trained to learn the key patterns of the distribution of outputs of the previous layer. Thus, the feature extractors are initialized in a region where they have already captured relevant information. Another method is using "short-cut" or "highway" connections (He et al., 2016; Huang et al., 2017; Srivastava et al.,

2015). In these methods, the feature extractor and classifier are updated in a more synchronized manner. Moreover, an auxiliary loss (Szegedy et al., 2015; Lu et al., 2019) can be added to the intermediate layers to bring the training signal closer to them and the layers below. Lu et al. (2019) show that for speech recognition tasks, even with 30,000 hours of training data, the training still suffers from the vanishing gradient problem.

In this paper, we propose to alleviate the vanishing gradient problem in a top-down manner. Instead of pre-training a feature extractor bottom-up layer by layer, we first train a good classifier, then freeze it, and then train the bottom layers with respect to the classifier.

To be more precise, we denote the loss function by $L$, the function defined by the frozen top layers by $T$ and the composition of these two functions by $L \circ T$. Training the bottom layers is equivalent to training a shallower network with the loss function $L \circ T$. Although the partial derivatives of the bottom layers may be small since $\partial L / \partial T$ can be small, this will only lead to slow training, not to under-fitting the feature extractor; this is because the classifier cannot be updated to overfit the outputs of the feature extractor or stop the training. Instead, the training terminates only when the feature extractor has fitted the current frozen classifier. Thus, the feature extractor cannot be left in an under-trained state. Moreover, the well-trained feature extractor has the potential of making the model generalize better. We summarize this argument as the *good classifier hypothesis*.

**The good classifier hypothesis.** We interpret the top layers of a neural network as a classifier and the bottom layers as a feature extractor. The vanishing gradient problem leads to the classifier being over-trained and the feature extractor under-trained. The under-trained feature extractor tends to capture the underlying patterns in the data less well. However, When trained with a frozen classifier, the feature extractor can fit to the classifier and becomes well-trained. If the classifier is good enough – that is, if it divides the space in a way such that the features that fit the partition reflect more about the essential patterns of the data – then the well-trained feature extractor is able to capture these patterns. As a result, the model will generalize better.

In this paper, we present a novel top-down training method based on the good classifier hypothesis, which we investigate with a set of speech recognition experiments. In these experiments we observe that the top-down approach leads to better generalization compared to conventional joint training. To our knowledge, this is the first time that this hypothesis is proposed. Also, to our knowledge, this is the first time that supervised top-down training is proposed and significant performance gains observed. Our findings have the potential to guide the development of new training algorithms and new transfer learning methods.

## 2 TRAINING FEATURE EXTRACTOR BASED ON A GOOD CLASSIFIER

In this section, we empirically justify the good classifier hypothesis. Before we give grounds for the hypothesis, we define the notion of a "good classifier". Here, the word "good" is a relative term. To better illustrate what is a good classifier, we first define *a cut of a neural network*.

**A cut of a neural network.** For a $n$-layer neural network, a cut $C_i$ is a partition of the layers $\boldsymbol{l} = (l_1, \ldots, l_i, l_{i+1}, \ldots, l_n)$ into two ordered sequences of layers $\boldsymbol{b} = (l_1, \ldots, l_i)$ and $\boldsymbol{t} = (l_{i+1}, \ldots, l_n)$, where $l_1$ denotes the bottom-most layer and $l_n$ denotes the top-most layer.

With this definition, we define *a good classifier*.

**A good classifier for a neural network.** For a $n$-layer neural network, we apply a cut $C_i$ to it. To the top layers $\boldsymbol{t}$ we assign a set of parameters and denote it as $\boldsymbol{t}_a$. We assign $\boldsymbol{t}$ another set of parameters and denote it as $\boldsymbol{t}_b$. If the features fit to $\boldsymbol{t}_a$ contain more information about the data pattern than features fit to $\boldsymbol{t}_b$, then $\boldsymbol{t}_a$ is a good classifier relative to $\boldsymbol{t}_b$.

By the good classifier hypothesis, the model trained with the frozen classifier $\boldsymbol{t}_a$ should generalize better than the model trained with the frozen classifier $\boldsymbol{t}_b$, since $\boldsymbol{t}_a$ will force the feature extractor to learn more about the underlying pattern. To verify this, we have conducted a series of end-to-end speech recognition experiments using the Wall Street Journal (WSJ: Paul & Baker, 1992) and CHiME-4 (Vincent et al., 2017; Barker et al., 2017) corpora. We train a network on the WSJ si284 dataset, a large vocabulary clean speech dataset. We construct a subset (25% size) of the si284 dataset by randomly sampling utterances from si284. Then we train a model on the si284 subset.

| CER results for clean (WSJ) and noisy (CHiME-4) training and test conditions | | | | | | |
|---|---|---|---|---|---|---|
| Training set | CER(valid) | CER(eval) | Sum of $|\Delta\theta|$ for each BLSTM layer | | | |
| si284 subset | dev92 | eval93 | BLSTM4 | BLSTM3 | BLSTM2 | BLSTM1 |
| Joint training | 20.3 | 17.5 | 41309 | 35512 | 32863 | 18798 |
| Frozen Classifier (Classifier trained on si284 subset) | 19.6 | 17.1 | _ | 42466 | 38985 | 22135 |
| Frozen Classifier (Classifier trained on full si284 ) | **15.6** | **12.5** | _ | 51279 | 47988 | 27492 |
| tr05_multi | dt05_multi | et05_real | BLSTM4 | BLSTM3 | BLSTM2 | BLSTM1 |
| Joint training | 42.0 | 32.3 | 46777 | 39982 | 39258 | 18602 |
| Frozen Classifier (Classifier trained on tr05_multi) | 41.5 | 31.0 | _ | 40052 | 37980 | 17093 |
| Frozen Classifier (Classifier trained on full si284 ) | **38.8** | **27.7** | _ | 66670 | 61078 | 29197 |

Table 1: CER for models trained on smaller clean (WSJ si284 subset) and noisy (CHiME tr05_multi) data. We view the softmax layer, the topmost projection layer and the topmost BLSTM layer as the classifier. Joint training means all layers are trained jointly. The next experiments are to freeze the classifier trained from the joint training then reinitialize and retrain the feature extractor. The final experiments are to freeze the classifier which trained on full si284 and train the feature extractors on si284 subset/tr05_multi. The right part of the table is the sum of the absolute values of the changes of all parameters of each BLSTM layer from epoch 1 to the epoch which has the lowest valid loss. $\Delta\theta$ denotes the change of a parameter.

We also train a model using CHiME-4 tr05_multi, a dataset containing WSJ utterances with real and simulated noise.

We assume the top layers of the model trained on the full WSJ si284 set is a good classifier, while the top layers of the models trained on si284 subset or on tr05_multi are poorer classifiers – since it is easier for a network trained on a relatively large, clean dataset (such as si284) to learn the underlying patterns in the data, compared with a network trained on the smaller (si284 subset) or noisier (tr05_multi) dataset. Therefore, if we can show that a good classifier (trained on si284) forces the feature extractor to learn useful patterns even from a small (si284 subset) or noisy dataset (CHiME-4), then this is evidence to support the good classifier hypothesis.

**Experimental setup.** We use Kaldi (Povey et al., 2011) to extract 40-dimension filterbank features with three pitch features, and the toolkit ESPnet (Watanabe et al., 2018) to implement the end-to-end models. We build bidirectional long short-term memory (BLSTM) (Hochreiter & Schmidhuber, 1997) connectionist temporal classification (CTC) models (Graves et al., 2006). All models have the same architecture. The network has 4 BLSTM layers. On top of each BLSTM layer is a projection layer, which is a linear layer with tanh activation functions. The top-most layer is a softmax layer. Thus, the model has 9 layers in total. Except for the softmax layer, each layer has 320 hidden units. The outputs are 26 characters, apostrophe, period, dash, space, noise, sos/eos tokens and some other special tokens, making 50 output labels in total. We used the adadelta (Zeiler, 2012) optimization algorithm. We used WSJ dev93 and CHiME-4 dt05_multi_isolated_1ch_track as validation sets for the models trained on WSJ and CHiME-4 respectively. The training stops if after 5 epochs there is no improvement upon the best validation loss.

Table 1 summarizes the experiment results in terms of character error rates (CER), and indicates that the feature extractor can be trained to fit a frozen classifier. Furthermore, it shows that the classifier trained on the full si284 dataset is a good classifier – with the frozen classifier trained on full si284, the model trained on the si284 subset and CHiME-4 has significant lower CERs. Lower CERs also indicate that the network learns more about the underlying pattern. Thus, the experiments of training with the good classifier show that to capture underlying patterns, the weights of the feature extractor

should have large changes. However, the vanishing gradient makes this almost impossible when all layers are trained jointly. These experiments support the good classifier hypothesis.

When the classifier trained on the si284 subset is frozen, the feature extractor become well-trained compared with the under-trained feature extractor in joint training. When trained with a frozen classifier from the joint training, the weights of the feature extractor have larger changes, indicating that it fits the classifier. For the CHiME-4 experiments, however, fixing the classifier from the joint training only makes the top layer of the feature extractor have a larger weight change. We suggest that the reason for this is that when all layers are trained on the noisy dataset jointly, the middle layers overfit the bottom-most layers much faster than the bottom-most layers are able to learn input features. Thus, at the end of training, the middle layers are also over-trained and should be viewed as part of the over-trained classifier. Only the bottom-most layers should be viewed as the under-trained feature extractor. One piece of evidence for this is if we compare joint training on tr05_multi and on si284, the changes of weights for the first BLSTM layer is comparable. However, for all other higher layers, the weights have larger changes if they are trained on tr05_multi, the noisy dataset. Our further experiments support this argument (section 5).

In these experiments, training with the frozen good classifier terminates after more epochs compared to joint training or the training with poorer classifiers. This is consistent with the assumption that when training with a frozen good classifier, the main drawback of potentially small partial derivatives is slowing down training. Additionally, when taking the classifier from the model trained on full si284, we only take the top-most BLSTM layer and the layers above it. We do not make the classifier deeper, since if we take and freeze more layers, the model will become very similar to the model trained on full si284. As a result, it would be questionable whether the gains in accuracy are from better training through the fixed good classifier or because the model is indistinguishable from the model trained on full si284. Finally, note that these experiments are conducted to support the good classifier hypothesis. The intention is not to perform transfer learning, although it is possible to develop a transfer learning method based on this approach.

## 3 TOP-DOWN TRAINING METHOD

The good classifier hypothesis states that if we have a good classifier, we can use it to guide the training of the lower feature extraction layers and the model will generalize better. We argue that when joint training converges, the top layers of a network make a potential good classifier – after convergence, the top layers will (further) overfit the bottom layers; before convergence, it is possible that the network does not learn much essential information and the top layers may not be well-trained yet. Hence we freeze the top layers at the time of convergence, then reinitialize and retrain the bottom layers. This *top-down training strategy* is presented as Algorithm 1.

---

**Algorithm 1** The top-down training strategy

---

1: For a $n$-layer neural network, train all layers jointly to obtain a trained network $A_n$
2: $i \leftarrow n, j \leftarrow i - 1$
3: **while** $i > 1$ **and** $j > 0$ **do**
4: $\quad j \leftarrow i - 1$
5: $\quad$ **while** $j > 0$ **do**
6: $\quad\quad$ Apply cut $C_j$ to $A_i$ to get classifier $\boldsymbol{t}$ and feature extractor $\boldsymbol{b}$
7: $\quad\quad$ Freeze $\boldsymbol{t}$; reinitialize and retrain $\boldsymbol{b}$ to obtain $A_j$
8: $\quad\quad$ **if** $A_j$ performs better on the valid set than $A_i$ **then**
9: $\quad\quad\quad$ $i \leftarrow j$
10: $\quad\quad\quad$ break
11: $\quad\quad$ **else**
12: $\quad\quad\quad$ $j \leftarrow j - 1$
13: $\quad\quad$ **end if**
14: $\quad$ **end while**
15: **end while**
16: **Output:** A model $A_k$ with the best validation performance

---

Table 2: CER for training on WSJ si284

| Model(WSJ si284) | dev_93 | eval_92 |
|---|---|---|
| Baseline 1 | 12.4 | 9.7 |
| + top-down training | 10.8 | 8.2 |
| + fix bottom layer | 13.1 | 10.5 |
| Baseline 2 | 12.6 | 10.4 |
| + top-down training | 10.6 | 8.5 |
| Baseline 3 | 12.6 | 10.2 |
| + top-down training | 11.5 | 8.9 |
| Dropout 0.2 | 11.0 | 8.7 |
| Dropout 0.5 | 9.6 | 7.6 |
| Dropout 0.7 | 11.1 | 8.9 |
| Dropout 0.5 | 9.6 | 7.6 |
| + top-down training | **8.2** | **6.3** |
| Previous work (Kim et al., 2017) | | |
| CTC | 11.5 | 9.0 |
| Attention (location) | 12.0 | 8.2 |
| CTC-Attention | 11.3 | 7.4 |

In this training strategy, if cut $C_j$ fails to improve validation performance, then retraining takes place using cut $C_{j-1}$. Since the top layers learn faster, it is possible the top layers with cut $C_j$ already overfit. However, the middle layers learn slower and they are less prone to overfitting. Thus, if we add more middle layers to the classifier, the classifier may overfit less. Furthermore, although with cut $C_{j-1}$, the feature extractor has less capacity, the classifier is more powerful. Thus, it reduces the burden on the feature extractor to learn complex features, making training easier. We observe this situation in the noisy speech recognition tasks.

If retraining with cut $C_j$ gives better validation performance, the top layers defined by $C_{j-1}$ may be a good classifier compared with the top layers defined by the same cut $C_{j-1}$ before the retraining with the cut $C_j$. Thus, we continue this training method from top to bottom.

The complexity of this method is $O(n)$, where $n$ is the number of the layers of the network. In our experiment, we execute this algorithm in a greedy manner. The inner while loop is only performed when $i = n$. After that, the algorithm goes down layer by layer, unless it reaches the bottom or the validation performance stops increasing. If retraining with the fixed classifier defined by cut $C_j$ does give an improvement on the validation set, instead of testing cut $C_{j-1}$, another possible solution is to choose the top layers at some epoch before convergence as the potential good classifier. However, this makes the computation more expensive and we have not conducted experiments on it.

## 4 TOP-DOWN TRAINING ON CLEAN DATASET

We test our top-down training method on the full WSJ si284 dataset. To reduce random factors, we train three baselines with different initialization. We apply our training method to all three baseline models. The architecture of the models is same as in section 2. Each model has 9 layers in total. To make the computation faster, we view each BLSTM layer and its following projection layer as "one layer". In this view, the network has 5 layers – one softmax layer and four BLSTM with its projection layers. For baseline 1, we also tested freezing the bottom-most BLSTM layer as well as its projection layer, then reinitialize and retrain all the top layers.

Table 2 demonstrates that top-down training reduces significantly the CER for each of the baseline models, further supporting the good classifier hypothesis. It indicates that in the joint training, when the classifier already has learned a reasonable partition, the feature extractor is usually under-trained. The additional experiment on baseline 1, indicates that if we freeze the bottom-most layer and retrain the top layers, the accuracy of the model drops, which is also consistent with our hypothesis – since the bottom layers are under-trained, fixing them and retraining the top layers is not helpful.

Figure 1 shows further training of the baseline does not bring any improvement. Rather, the validation loss is increasing during the long training, indicating that instead of making the feature

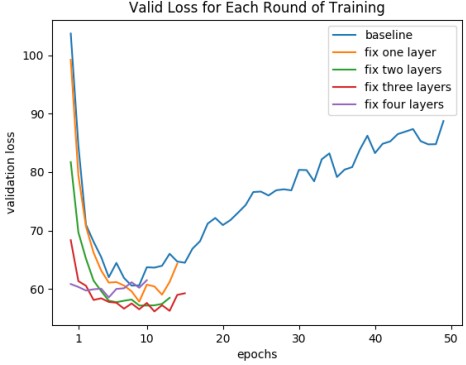

Figure 1: Validation loss

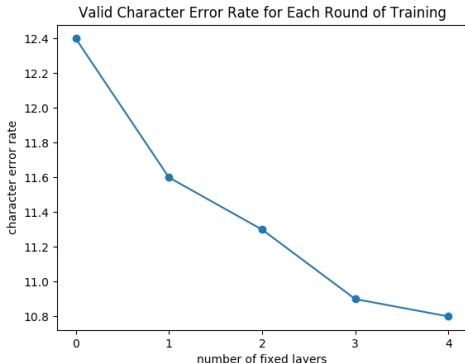

Figure 2: Validation character error rate

extractor well-trained and learn more about the data pattern, it makes the classifier further overfit the feature extractor. Furthermore, compared to the baseline, some rounds of retraining take more epochs to reach the lowest validation loss, confirming that the small partial derivatives of the bottom layers may make the training slow but cannot prevent the feature extractor from being well-trained. Figure 2 shows that in each round of the top-down training, we obtain better feature extractors.

Note that although our network has an identical architecture to that used by Kim et al. (2017), our baseline models have higher CER. We assume that this may be caused by other inconsistent training settings, such as the input features and the output labels. To ensure we have valid baselines and our training method can be combined with other training strategies, such as dropout (Srivastava et al., 2014), we use dropout during training and then combined it with our method. We also built a deep convolutional neural network (CNN)-BLSTM CTC model to get a better baseline. The architecture and results for CNN-BLSTM are in the appendix.

Using the same initial weights as the initial weights of baseline 3 (where our method brings the least improvement, although the improvement is also significant), we train three models with different dropout probabilities, with dropout probability $0.5$ yielding the best model, followed by top-down training (still using dropout probability $0.5$).

The results in Table 2 show that when combined with dropout, top-down training brings large gains in accuracy over both the baseline and the model trained with dropout ($38\%$ and $17\%$ relative gain, respectively). The combination of these two training methods performs impressively, with the CTC model outperforming the joint CTC-attention model significantly ($15\%$ relative gain).

The top-down training can be viewed as shrinking the size of network "vertically", as the fixed classifier can be viewed as part of the loss function. Dropout can be regarded as an approach to reduces the size of the network horizontally, by dropping nodes in each layer according to a probability. Combining these methods results in a network which at training time is effectively shallower and narrower, while retaining a deep and wide network at run-time.

## 5   TOP DOWN TRAINING ON NOISY DATA

To test if our training method is valid using noisy data, we run experiments using the CHiME-4 data set, using the same architecture as in section 2. We again investigated combining top-down training with dropout. All the models had the same initial weights. Results are summarized in table 3.

Compared with the clean WSJ experiment, top-down training brings smaller relative improvement ($5\%$ vs. $13\%$). For this experiment, in the first round of retraining, we needed to view the softmax layer and the top two BLSTM layers with their projection layers as the fixed classifier to get performance gain on the validation set. By contrast, for WSJ, in the first round of retraining, fixing the softmax layer alone was sufficient. This is indicative of the difficulty of learning underlying patterns from the noisy data. Thus, overfitting of the top layers contributes more to the convergence of the model. To have a good classifier rather than having a classifier that overfits, we need to add more

Table 3: Character error rate (CER) on CHiME-4

| tr05 + si284 | dt_05 (CER) | et_05 (CER) |
|---|---|---|
| baseline | 25.2 | 36.0 |
| +top-down training | 23.4 | 34.2 |
| dropout 0.2 | 22.8 | 34.7 |
| dropout 0.5 | 19.9 | 30.8 |
| dropout 0.7 | 20.3 | 30.3 |
| dropout 0.5 + top-down training | **18.5** | **28.7** |
| dropout 0.5 + fix the bottom layer | 20.6 | 31.4 |

middle layers to the classifier, which are less prone to overfitting. Furthermore, with more layers, the classifier has increased capacity, freeing the feature extractor from needing to learn more complex features from the noisy data.

Dropout reduces CER significantly. It indicates that models trained with dropout learn more about the data patterns and the top layers of these models make good classifiers compared to the poorer classifiers of the baseline model trained with joint training. Thus, in the first round of retraining of the model trained with dropout (probability 0.5), fixing the softmax layer and the topmost BLSTM layer with its projection layer yields validation accuracy gains. Finally, dropout (probability 0.5) combined with top-down training resulting in a relative gain of 7%.

Even for the model trained with dropout, if we fix the bottom-most BLSTM layer with its projection layer and retrain the top layers, the accuracy of the model drops. This observation also supports the argument that due to the vanishing gradient, the bottom layer is under-trained and it does not capture much underlying patterns. So, retraining with fixed bottom layers does not help.

## 6 DISCUSSION AND CONCLUSION

In the clean WSJ experiments, when combined with dropout, top-down training enables a CTC model to significantly outperform a CTC-attention model with the same encoder architecture. The CTC results could be improved through techniques such as a more complex network architecture, curriculum learning, data perturbation, and language model fusion. We do not use these approaches in order to ensure precise comparisons on the networks trained with top-down training and with conventional joint training. The main goal of this work is to show that the top-down training method alleviates the problem of under-trained feature extractors, which is caused by vanishing gradients. Testing our training method on more elaborate models and on different datasets and domains is left as a further work.

In this work, we show that top-down training can be combined with dropout, resulting in substantial accuracy gains. Moreover, top-down training does not conflict with bottom-up pretraining or the residual/highway connections, leading to some interesting future directions.

Currently, we propose fixing the classifier and retraining the feature extractor. An alternative approach is to halt the training of the classifier instead of fixing it. Since our training method can enable the feature extractor to be better trained, adjusting the classifier according to the better-trained feature extractor may bring further gains. Thus, after top-down training, we could freeze the feature extractor and continue to train or re-train the classifier.

In top-down training, we view the fixed classifier as part of the loss function. The supervision signal is still the true label. Alternatively, we could compute the pseudo-inverses of the weight matrices of the classifier and use these inverses as well as the true label to compute the expected input for the classifier with the current true label, using the expected input as the label for the feature extractor. We could interpret this process as a form of label smoothing.

Obtaining a good classifier is essential to top-down training. To stabilize the approach, it is possible to train several networks with different initializations, resulting in a set of trained classifiers with different weights. In this case, training the feature extractors can be interpreted as a form of multi-task learning with each trained classifier in the set.

Although many popular optimization algorithms dynamically adjust the learning rate (Qian, 1999; Duchi et al., 2011; Zeiler, 2012; Kingma & Ba, 2014; Dozat, 2016), they do not explicitly consider the information of the position of the layers. Our results indicate that this information is crucial in adjusting the learning rate (the frozen classifier has a learning rate 0).

Also, by using the classifier trained on WSJ to guide the training of feature extractors on CHiME-4, we observe significant performance gains. Therefore, it is possible that by fitting a classifier trained on a large clean dataset, the feature extractor is forced to learn denoising. This observation suggests a route to design new transfer learning methods from clean datasets to noisy datasets, which is important for robust speech recognition.

In summary, we observe that the vanishing gradient problem causes the classifier to be over-trained and the feature extractor to be under-trained. Top-down training alleviates this problem and results in better model generalization.

## 7 RELATED WORK

To better train DNNs, layer-wise pre-training has been widely studied (Hinton et al., 2006; Bengio et al., 2007; Larochelle et al., 2009; Erhan et al., 2010). In this method, each layer learns to capture the most relevant variations of the distribution of the input or the outputs of the previous layer. Finally, fine-tuning is applied for the classification tasks. In terms of the feature extractor-classifier, pre-training restricts the initialization of the feature extractors to a good region, while top-down training creates good classifiers and adjusts feature extractors based on good classifiers. These two methods can be easily combined and may lead to further improvements. Supervised pre-training is also studied, but in general this method does not outperform unsupervised pre-training (Bengio et al., 2007; Larochelle et al., 2009).

Recently, bottom-up layer-wise training (Hettinger et al., 2017; Belilovsky et al., 2018; Zhao et al., 2018) has been investigated in which the network is trained in a bottom-up way and no fine-tuning is applied. The training method of Hettinger et al. (2017) yields worse results compared to conventional training on their DNN experiments on the MNIST dataset (LeCun, 1998). Belilovsky et al. (2018) report that by performing bottom-up layer-wise training using auxiliary networks, they build networks that outperform AlexNet (Krizhevsky et al., 2012) and VGG (Simonyan & Zisserman, 2014) on ImageNet (Deng et al., 2009). However, if the networks with auxiliary networks are trained conventionally, further gains are obtained. Zhao et al. (2018) train all layers jointly and retrain the topmost layer multiple times to obtain a large ensemble of networks. It is unclear how efficient this method is for each single training.

Architectures with skip connections, such as ResNets (He et al., 2016), dense nets (Huang et al., 2017) and highway networks (Srivastava et al., 2015) also alleviate the vanishing gradient problem. Through the skip connections, the error signal from the higher layers can be directly back-propagated to the lower layers, without passing through the non-linear intermediate layers. Alternatively the supervision signal can be routed directly to the lower layers. Szegedy et al. (2015) propose adding an auxiliary classifier to the intermediate layers. Lu et al. (2019) propose to add the KL divergence between the output of the top layer and the output of a middle layer as an auxiliary loss. They obtain word error rate reductions after training on 30,000 hours of speech data, which indicates that even with large amounts of training data, the vanishing gradient problem is still significant. Top-down training alleviates the vanishing gradient problem from a different angle and can be integrated with these methods.

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

# A    Top-down training for CNN-BLSTM Models

We test our method in training a deep CNN-BLSTM CTC model. The purpose of this experiment is to verify our training method works for a hybrid CNN-BLSTM architecture. The model has 6 BLSTM layers. Each BLSTM layer is followed by a linear layer with tanh activation. Each BLSTM layer, as well as each linear layer, has 320 hidden units. The bottom-most BLSTM layer is on top of CNN layers. The architecture of the CNNs is in Table 4.

Table 4: The CNN architecture for the CNN-BLSTM model

|         | in_channel | out_channel | kernel size | stride |
|---------|-----------|-------------|-------------|--------|
| conv    | 1         | 64          | $3 \times 3$ | 1      |
| conv    | 64        | 64          | $3 \times 3$ | 1      |
| maxpool |           |             | $2 \times 2$ | 2      |
| conv    | 64        | 128         | $3 \times 3$ | 1      |
| conv    | 128       | 128         | $3 \times 3$ | 1      |
| maxpool |           |             | $2 \times 2$ | 2      |

Table 5: Character error rate (CER) for WSJ of the CNN-BLSTM model

| Model(WSJ si284)              | dev_93 | eval_92 |
|-------------------------------|--------|---------|
| CNN-BLSTM                     | 10.9   | 8.2     |
| + top-down training           | 9.7    | 7.3     |
| Dropout 0.2                   | 9.9    | 7.5     |
| Dropout 0.5                   | 10.3   | 7.6     |
| Dropout 0.7                   | 11.6   | 9.3     |
| Dropout 0.2                   | 9.9    | 7.5     |
| + top-down training           | 8.4    | **6.3** |
| BLSTM                         | 12.6   | 10.2    |
| + dropout 0.5 + top-down training | **8.2** | **6.3** |
| Previous work (Kim et al., 2017) |     |         |
| CTC                           | 11.5   | 9.0     |
| Attention (location)          | 12.0   | 8.2     |
| CTC-Attention                 | 11.3   | 7.4     |

Table 5 demonstrates our training method is still successful in training a much deeper hybrid CNN-BLSTM model. Furthermore, in this set of experiments, applying our training method solely outperforms applying dropout solely. Moreover, with dropout (probability 0.2) combined with top-down training, further gain is obtained. In these experiments, the top-down training method for the CNN-BLSTM stops at the second top-most BLSTM layer. Further accuracy gains may obtained if we execute the inner loop in Algorithm 1. We also tested from the joint trained CNN-BLSTM model (with dropout probability 0.2), fixing all BLSTM layers together and retraining all the CNN layers in the same time. No further layer-wise cuts or retraining were applied to the CNN layers yet. With a $9.7\%$ valid CER and a $7.4\%$ eval CER, the model with retrained CNN layers outperforms the baseline, but has inferior results compared to the model trained with top-down training from the top-most softmax layer.

One interesting observation is although the baseline CNN-BLSTM model has lower CERs compared to the baseline BLSTM model, applying the combination of dropout and top-down training to both models yields similar CERs. CNNs are effective in extracting features, making the CNN-BLSTM model outperforms the BLSTM model. Our top-down training method enables the network learn the patterns of the data well and extract appropriate features. Therefore, after applying the top-down training method (with dropout), the pure BLSTM model has a similar performance to the CNN-BLSTM model.

## B    PRELIMINARY STUDIES ON LANGUAGE MODEL

To verify the effectiveness of our top-down training method is not limited to the domain of speech recognition, we tested our method for a language modelling task. We train a two-layer LSTM language model using the transcriptions of WSJ si284 dataset. Each LSTM layer has 256 hidden units. The transcriptions contain 644,177 words. The vocabulary size is 20,000. The out of vocabulary rate is 2.45%. The optimisation algorithm is stochastic gradient descent with a learning rate 1.0. We also tested to use Adam (Kingma & Ba, 2014) and adadelta (Zeiler, 2012) as the optimisation algorithm, but they give inferior results. Dropout (probability 0.5) is applied. We use the transcriptions of WSJ dev_93 and WSJ eval_92 as the validation/test set. They contain 8,334 and 5,700 words respectively.

Table 6: Perplexity of the 2-layer LSTM model.

|  | dev_93 (perplexity) | eval_92 (perplexity) |
|---|---|---|
| Joint Training | 84.0 | 81.2 |
| Top-down Training | 82.6 | 79.0 |

From Table 6, our preliminary study shows the proposed training method is helpful in a natural language processing task. Further studies on more advanced models should be conducted on benchmark language model datasets.

