# OpenReview forum: "Top-down training for neural networks"
_ICLR.cc/2020/Conference — Reject_

### Official Review · AnonReviewer2 · 2019-10-23
**Official Blind Review #2**

**Rating:** 3

**Review:**

This work proposed a mechanism to freeze top layers after supervised pre-training, and re-initialize and retrain the bottom layers. For a model with n layers, when a separation index i is specified, the approach define layer 1~i as bottom layers and i+1~n as top layers. The proposed process enumerate all i from 1 to n-1, compute resulting validation errors respectively, and then pick the i with lowest validation error. The algorithm exhibited significant improvement on WSJ and some minor improvement on CHiME-4.

This work provides some new insight for training ASRs and the observations provide further data points for understanding the training behavior. The layer freezing trick however is relatively well-known, and thus leaving the novelty of the proposed idea to be limited at what layers they choose to freeze.

In algorithm 1 it describes the mechanism as having two loops while it really only needs one loop. The author mentioned they used a simplified version later in the text, and I’ll suggest to update the algorithm block to make it clearer.

**Experience Assessment:**

I have published in this field for several years.

**Review Assessment: Checking Correctness Of Derivations And Theory:**

I carefully checked the derivations and theory.

**Review Assessment: Checking Correctness Of Experiments:**

I carefully checked the experiments.

**Review Assessment: Thoroughness In Paper Reading:**

I read the paper at least twice and used my best judgement in assessing the paper.

---

> ### Author Response · Authors · 2019-11-14
> **Comments about the novelty of the paper**
>
> Comments about the novelty of the paper
>
> Thank you for your comments! We believe our work has many new findings and contributions.
> Firstly, to our knowledge, this is the first time that layer-wise training is explored in a top-down manner and it is the first time that layer-wise training (not layer-wise pre-training; there is no joint fine-tuning) outperforms joint training significantly.
>
> There are quite a few recent papers working on the layer-wise training [1,2,3]. However, all of them investigate this approach in a bottom-up method and do not get considerable improvements over the joint training. In fact, the authors of [2] state that their work is the first time that such training approach has comparable results to the joint training on a large scale dataset. However, we investigate it in a top-down manner and the top-down training significantly surpasses the joint training.
>
> Secondly, we have an insightful analysis on the reason of the effectiveness of the top-down training, from the perspective of the vanishing gradient and training classifier-feature extractor. To our knowledge, we believe it is the first time that such an analysis is made. Also, our analysis gives the reasons for why layer-wise training in a bottom-up way is in general hard.
>
> Thirdly, many popular optimizers, such as Adadelta and Adam, dynamically adjust the learning rate. However, we find that the way of adjusting learning rate provided by these optimizers is not optimal. They have inferior results compared with freezing the upper layers. Thus, our findings lead to a new direction of designing optimization algorithms.
>
> In summary, the novelty of this paper is not limited to choosing which layer to freeze. It is the first time that layer-wise training outperforms the joint training significantly. We also have an insightful analysis on the proposed training method, which explains why the top-down manner is beneficial and the bottom-up method is hard. From the proposed method, a new optimization algorithm may be developed.
>
> [1] Chris Hettinger, Tanner Christensen, Ben Ehlert, Jeffrey Humpherys, Tyler Jarvis, and Sean Wade. Forward thinking: Building and training neural networks one layer at a time. arXiv preprint arXiv:1706.02480, 2017.
> [2] Eugene Belilovsky, Michael Eickenberg, and Edouard Oyallon. Shallow learning for deep networks. openreview, 2018. URL https://openreview.net/forum?id=r1Gsk3R9Fm.
> [3] Kaikai Zhao, Tetsu Matsukawa, and Einoshin Suzuki. Retraining: a simple way to improve the ensemble accuracy of deep neural networks for image classification. In 2018 24th international conference on pattern recognition (ICPR), pp. 860–867. IEEE, 2018.

---

### Official Review · AnonReviewer1 · 2019-10-23
**Official Blind Review #1**

**Rating:** 3

**Review:**


=========================
Update review
After reading the authors response I would like to keep my score as is.
I still see many unclear statements, and most importantly I feel that more analysis of the proposed method should have been done here.
=========================

This paper proposed a Top-Down method for neural networks training based on the good classifier hypothesis. In other words, after obtaining a classifier that performs well on the test set, keep fine-tuning / re-learning the data representation.
The authors provide character error rate results for the task of Automatic Speech Recognition using WSJ and CHiME-4 datasets.

Although being an interesting research idea, several issues in this paper make it not yet ready for publication at ICLR.

First, the paper is poorly written; there are many claims the authors are making without providing experiments/proofs/citations.
For example: "...since the feature extractor learns more slowly, then potentially the classifier may overfit the feature extractor before the feature extractor is able to learn much about the underlying pattern of the data...".
Or: "...We suggest that the reason for this is that when all layers are trained on the noisy dataset jointly, the middle layers overfit the bottom-most layers much faster than the bottom-most layers are able to learn input features..."

Next, since there is no theoretical/mathematical explanation of the proposed approach, I expect the authors to run an analysis on the results to better understand the effect of using such an approach. For instance, under which settings this method is most efficient? In what layer should  I start the fine-tuning? Is it better to reinitialize the bottom layers or fine-tune them? Does the proposed approach applicable to different domains? i.e. vision/nlp/other speech/signal processing tasks?  Does the proposed approach applicable to different models or only for the proposed one?

Lastly, although it is not the main point in this paper since all results are reported on ASR, did the authors tried to compute WERs too? That way, people can compare results with other ASR models. The baseline seems relatively weak, at least in Table 1.

Minor comments:
The complexity of the algorithm is written to be O(n). However, this assumes training the model takes O(1) or did I miss something?
Can the authors provide more details/insights regarding the delta differences in Table 1? Did the authors use the same initializations? Did the authors try different ones?

**Experience Assessment:**

I have read many papers in this area.

**Review Assessment: Checking Correctness Of Derivations And Theory:**

N/A

**Review Assessment: Checking Correctness Of Experiments:**

I carefully checked the experiments.

**Review Assessment: Thoroughness In Paper Reading:**

N/A

---

> ### Author Response · Authors · 2019-11-14
> **We suppose almost all the questions are answered in the paper (part 1)**
>
> Thank you for your comments. Indeed, we find most questions are already addressed by the paper. We pinpoint the sections of the paper which address these questions. Please find below our step-by-step answers to your comments.
>
> 1. First, the paper is poorly written; there are many claims the authors are making without providing experiments/proofs/citations.
>
> We make these claims based on carefully designed experiments (section 2 of the paper). We justify our claims through showing the weight changes during training and the CERs of each model. The change of the weights indicates if a layer is “learning”. If all layers are trained jointly, top layers have larger weight changes and the model has high CERs. If trained with frozen top layers, compared to the jointly trained models, the lower layers have larger weight changes the model has much lower CERs. Thus, it shows in the joint training the top layers overfits the lower layers and the lower layer should be further trained.
>
> 1.(b) For example: "...since the feature extractor learns more slowly, then potentially the classifier may overfit the feature extractor before the feature extractor is able to learn much about the underlying pattern of the data...". Or: "...We suggest that the reason for this is that when all layers are trained on the noisy dataset jointly, the middle layers overfit the bottom-most layers much faster than the bottom-most layers are able to learn input features..."
> Please find our detailed answer in the bottom.
>
> 2. Next, since there is no theoretical/mathematical explanation of the proposed approach, I expect the authors to run an analysis on the results to better understand the effect of using such an approach.
>
> We believe the analysis in section 2 justifies the proposed approach. Also, we have detailed analysis of our experiments in section 4 and section 5.
>
> 3. For instance, under which settings this method is most efficient? In what layer should I start the fine-tuning? Is it better to reinitialize the bottom layers or fine-tune them?
>
> All these questions can be answered by Algorithm 1 in section 3, and the analysis part in section 4 and section 5. The top-down training starts from freezing the topmost layer. The bottom layers should be reinitialized.
>
> 4. Does the proposed approach applicable to different domains? i.e. vision/nlp/other speech/signal processing tasks?
>
> In the main part of the paper we show it works for both clean/noisy speech data. In the appendix, we show our initial results for training a language model.
>
> 5. Does the proposed approach applicable to different models or only for the proposed one?
>
> In the appendix, we show the results for CNN-BLSTM. We also have results for encoder-decoder models on TIMIT. The results are in phone error rate and compared with other end-to-end models.
> BiGRU encoder-decoder [5]                     18.7
> Segmental RNN [6]                                    20.5
> BLTM CTC[7]                                               24.6
> BLSTM encoder-decoder (ours)                19.1
> BLSTM Encoder-decoder + top-down      18.4 (we just tried to freeze the decoder)
>
> 6. did the authors tried to compute WERs too
>
> We state the reasons in the first paragraph of section 6. The goal of this paper is to show the effectiveness of the proposed training method. Thus, we need exact fair comparisons. If we report WERs then possibly we need to decode with a language model. However, in this case, we will have an extra component and it may make blur the comparisons.
>
> 7. The baseline seems relatively weak, at least in Table 1.
>
> As stated in section 2, the experimental results in Table 1 is to support the good classifier hypothesis. The purpose of these experiments is not to show some good CERs.  Indeed, for Table 1, the datasets are a subset of the full WSJ and CHiME-4 without data augmentation. Thus, with the current models, it is hard to get very low CERs.
> We show that the proposed methods can lead to good CERs using full WSJ and CHiME-4 with data augmentation. For WSJ eval 92, here are more results (CER) from previous works:
>
> BRDNN CTC [1]                                 10.0
> BLSTM CTC [2]                                   9.2
> BLSTM CTC [3]                                   9.0
> Encoder-Decoder [3]                        8.2
> CTC- Encoder-Decoder [3]               7.4
> Encoder-Decoder + TwinNet [4]      6.2
> Ours (BLSTM CTC)                             6.3
>
> Thus, we suppose we have a very strong number for our CTC models. Compared to the encoder-decoder models which have more capacity and which are more flexible, the CTC model trained with the proposed method have better/comparable results.

---

> > ### Author Response · Authors · 2019-11-14
> > **We suppose almost all the questions are answered in the paper (part 2)**
> >
> > 8. The complexity of the algorithm is written to be O(n). However, this assumes training the model takes O(1) or did I miss something?
> >
> > Here we show the complexity of the top-down training method. The complexity of training the model is decided by the model, not the proposed training method. Thus, we do not consider the complexity of training the model.
> >
> > 9. Can the authors provide more details/insights regarding the delta differences in Table 1? Did the authors use the same initializations? Did the authors try different ones?
> >
> > The delta differences are the changes of the weights. The change of the weights indicates if a layer is “learning”. If all layers are trained jointly, top layers have larger weight changes and the model has high CERs. If trained with frozen top layers, compared to the jointly trained models, the lower layers have larger weight changes the model has much lower CERs. Thus, it shows in the joint training the top layers overfits the lower layers and the lower layer should be further trained.
> >
> > We do not use the same initialization. Also, random initialization is not a critical factor here, since it is almost impossible to get over 20% CER (which is achieved by frozen the topmost layer in table 1) reduction by just trying different random initializations. Furthermore, to preclude the random initialization factor, in our WSJ experiments, we build three baselines with different random initialization.
> >
> > [1] Hannun, Awni Y., et al. "First-pass large vocabulary continuous speech recognition using bi-directional recurrent dnns." arXiv preprint arXiv:1408.2873 (2014).
> > [2] Graves, Alex, and Navdeep Jaitly. "Towards end-to-end speech recognition with recurrent neural networks." International conference on machine learning. 2014.
> > [3] Kim, Suyoun, Takaaki Hori, and Shinji Watanabe. "Joint CTC-attention based end-to-end speech recognition using multi-task learning." 2017 IEEE international conference on acoustics, speech and signal processing (ICASSP). IEEE, 2017.
> > [4] Serdyuk, D., Ke, N., Sordoni, A., Trischler, A., Pal, C. and Bengio, Y. “TWIN NETWORKS: MATCHING THE FUTURE FOR SEQUENCE GENERATION”. International Conference on Learning Representations. 2018.
> > [5] Chorowski, Jan K., et al. "Attention-based models for speech recognition." Advances in neural information processing systems. 2015.
> > [6] Lu, Liang, et al. "Segmental Recurrent Neural Networks for End-to-End Speech Recognition." Interspeech 2016 (2016): 385-389.
> > [7] Fernández, Santiago, Alex Graves, and Jürgen Schmidhuber. "Phoneme recognition in TIMIT with BLSTM-CTC." arXiv preprint arXiv:0804.3269 (2008).

---

> > ### Author Response · Authors · 2019-11-14
> > **Detailed answers for question 1(b).**
> >
> > The description of the experiments is in the first paragraph of page 3 (under Table 1):
> >
> > “We assume the top layers of the model trained on the full WSJ si284 set is a good classifier, while the top layers of the models trained on si284 subset or on tr05 multi are poorer classifiers – since it is easier for a network trained on a relatively large, clean dataset (such as si284) to learn the underlying patterns in the data, compared with a network trained on the smaller (si284 subset) or noisier (tr05 multi) dataset. Therefore, if we can show that a good classifier (trained on si284) forces the feature extractor to learn useful patterns even from a small (si284 subset) or noisy dataset (CHiME-4), then this is evidence to support the good classifier hypothesis”
> >
> > We have experimental results support “since the feature extractor learns more slowly, then potentially the classifier may overfit the feature extractor before the feature extractor is able to learn much about the underlying pattern of the data”
> >
> > The evidence is stated in the first paragraph of page 4:
> >  “When trained with a frozen classifier from the joint training, the weights of the feature extractor have larger changes, indicating that it fits the classifier. For the CHiME-4 experiments, however, fixing the classifier from the joint training only makes the top layer of the feature extractor have a larger weight change.”
> >
> > As shown in Table 1, at the end of training, in the joint training case, the weights of the upper layers have much larger changes than the bottom layers, which indicates they learn faster. After the joint training, if the topmost layer is frozen and the bottom layers are reinitialized and retrained, for the subset of WSJ, the bottom layers’ weights have larger changes compared to the case of joint training.
> > The model trained with frozen topmost layer has lower CER. Thus, it indicates in the joint training, the unwanted converge is caused by the overfitting top layers.
> >
> > Also, if trained with a frozen classifier which is trained on full WSJ, then compared with models trained on subset of WSJ/CHiME-4, the lower layers’ weights have significantly larger changes and the models have considerably lower CER.
> >
> > “We suggest that the reason for this is that when all layers are trained on the noisy dataset jointly, the middle layers overfit the bottom-most layers much faster than the bottom-most layers are able to learn input features”
> >
> > The evidence is stated in the first paragraph of page 4:
> > “One piece of evidence for this is if we compare joint training on tr05 multi and
> > on si284, the changes of weights for the first BLSTM layer is comparable. However, for all other
> > higher layers, the weights have larger changes if they are trained on tr05 multi, the noisy dataset.”
> > Our further experiments support this argument (section 5).
> >
> > In section 5, we find for CHiME-4, it is more beneficial to start the top-down training by freezing the layers from the middle layer to the topmost layer, rather than just freezing the topmost layer.

---

### Official Review · AnonReviewer4 · 2019-11-26
**Official Blind Review #4**

**Rating:** 3

**Review:**

This paper studies the common experimental finding that low level features trained end-to-end in a deep model converge (get "locked in place") earlier than higher level features, which may result in problematic undertraining. The focus of the study is not on skip connections, but really on getting adequate training in deeper networks. They posit a "good classifier hypothesis" where, once a deep network converged, they fix the top layers (the "good classifier") and train only the lower ones. They propose a "top-down training strategy" to search where to make the cut for the "top layers" of the "good classifier", based on the validation set.


 (+) The experimental results seem encouraging and supporting the author's claim (consistently improve over baseline on WSJ and CHiME-4).
 (-) No WER (not even without a language model) results on WSJ make it harder to (i) compare to other work (is it just that in this case the authors didn't optimize properly in the first place?), (ii) compare the relative gains between with and without the method in WER.
 (-) For an experimental (no theorem) optimization paper, there should be experiments on at least another domain. And in particular one would have expected more analysis of the experimental optimization results.
 (-) (minor) There is no discussion of the link with target propagation or other synthetic gradients.

Overall, I think this could be an interesting paper, but more work is needed to prove the effectiveness of the method, and to analyze experimentally in more details some of the claims from this paper.

**Experience Assessment:**

I have published in this field for several years.

**Review Assessment: Checking Correctness Of Derivations And Theory:**

N/A

**Review Assessment: Checking Correctness Of Experiments:**

I assessed the sensibility of the experiments.

**Review Assessment: Thoroughness In Paper Reading:**

I read the paper at least twice and used my best judgement in assessing the paper.

---

### Public Comment · ~Huaxin_Song1 · 2019-10-15
**Which layers learn faster? Lower or deeper?**

Hello, I really enjoy reading your paper and it is insightful. In your abstract, you mentioned the top layers (closer to the output) in the network learning faster when compared with lower layers closer to the input. Could you explain more about why the gradient vanishing problem leads to faster learning in top layers?

I read a paper [1] saying that the lower layers of the model can be seen to converge within a few epochs. However, the upper layers only develop after a considerable number of epochs (40-50), demonstrating the need to let the models train until fully converged.  The authors of [1] seem to have different ideas.

[1] Zeiler M D, Fergus R. Visualizing and understanding convolutional networks[C]//European conference on computer vision. Springer, Cham, 2014: 818-833.

---

> ### Author Response · Authors · 2019-10-15
> **"Faster" here refers to the speed of fitting other layers, not the speed of learning features**
>
> Hi, thank you very much for your interests and your comments. I think there is no contradicts between our statement and [1]. We need to carefully define what the word “fast” means. In [1], it means the speed of learning useful features. In our statement, it means the speed of fitting other layers. Thus, it takes more epochs for the top layers to learn useful features; while in the same time, the top layers fit the bottom layers much faster than the bottom layers fit the top layers. In this work, we stop training the top layers when they begin to overfit the bottom layers; we are not saying the top layers learn useful features earlier, so we stop training it earlier. Actually, we stop the training of the top layer until the joint training converges.
>
> When we define the term “fast” in the scope of fitting other layers, in our paper, we did two experiments to show that after convergence, freezing the bottom (most) layer and retrain the top layers does not help.  In fact, we found the experiments in [2] further show that freezing the bottom layers does not help in general.
>
> [1] Zeiler M D, Fergus R. Visualizing and understanding convolutional networks[C]//European conference on computer vision. Springer, Cham, 2014: 818-833.
>
> [2] Yosinski, Jason, et al. "How transferable are features in deep neural networks?." Advances in neural information processing systems. 2014.

---

### Public Comment · ~TING_TING_SUN1 · 2019-10-28
**Some problems about the algorithm**

The upper layers of a *well trained* neural network can be only activated by some specific patterns.

However, during the training process of top-down training setting, the top layers are frozen while the bottom layers are trained with random initialization. It may lead to a problem that in forward pass of early stage the feature maps in the bottom layers are noise due to random initialization and they may not pass the *relu function* because top layers can be only activated by specific patterns. It means little signal can be passed in the upper layers and it may hurt the training process.

I am curious whether the authors encounter this situation because there seems no guarantee to avoid this problem in the algorithm.

---

> ### Author Response · Authors · 2019-10-28
> **A problem of ReLU activation in genearl, not a problem of the top-down training**
>
> Hi, thank you very much for your interets in this work and the comments. If the word "activated" means the output of ReLU is larger than $0$, I suppose it is not a problem for this training algorithm. Rather, it is a problem of the ReLU activation itself.
>
> In an abstract manner, for one hidden unit with paramter $w$, when $w$ is not frozen, although $w$ changes dynamically during training, in the current pass, still, this unit "can be only activated by specific patterns".
>
> In a more detailed way, for one hidden unit with paramter $w$, if the distribution of the input $x$ is symmetry through the origin (which is in general assumed or achieved by normalization), then the chance of $wx$ is larger than $0$ is always 50%, no matther the value of $w$ or if $w$ is frozen. If $w$ is not frozen and the sign of $w$ flips during training, then if the sign of $x$ does not change, whether $wx$ is larger than 0 also flips. However, statistically, the chance of if $wx$ is larger than $0$ remains 50% (if the distribution of $x$ also remains symmetry through the origin). Thus, whether $w$ is frozen or not, the chance of whether $x$ will have a non-zero partial derivative is always 50%.
>
> In terms of experimental results, our algorithm works for CNN-BLSTM models (taking appendix A for an example). We will do more experiments for pure CNN models.

---

### Decision · Program_Chairs · 2019-12-19

**Decision:**

Reject

**Comment:**

The paper proposes a top-down approach to train deep neural networks -- freezing top layers after supervised pre-training, then re-initializing and retraining the bottom layers. As mentioned by all the reviewers, the novelty is on the low side. The paper is purely experimental (no theory), and the experimental section is currently too weak. In particular:
- Experiments on different domains should be performed.
- Different models should be evaluated.
- Ablation experiments should be performed to understand better under which conditions the proposed approach works.
- For speech recognition, WER should be reported - even if it is without a LM - such that one can compare with existing work.